# Evaluation of Skin Penetration of Fluorescent Dissolved Formulations Using Confocal Laser Scanning Microscopy

**DOI:** 10.3390/pharmaceutics17121534

**Published:** 2025-11-28

**Authors:** Yoshihiro Oaku, Toshinari Kuwae, Takeshi Misono, Taku Ogura, Akinari Abe

**Affiliations:** 1Frontier Research Center, Taisho Pharmaceutical Co., Ltd., 3-24-1 Takada, Toshima-ku, Tokyo 170-8633, Japan; y-oaku@taisho.co.jp; 2Nikko Chemicals Co., Ltd., 3-24-3 Hasune, Itabashi-ku, Tokyo 174-0046, Japan; kuwatosh@nikkolgroup.com (T.K.); misotake@nikkolgroup.com (T.M.); taku_ogura@rs.tus.ac.jp (T.O.); 3Self-Medication Research Center, Research Headquarters, Taisho Pharmaceutical Co., Ltd., 1-403 Yoshino-cho, Kita-ku, Saitama 331-9530, Japan

**Keywords:** skin penetration, trans-stratum corneum route, trans-follicular route, confocal laser scanning microscopy, three-dimensional visualization

## Abstract

**Background/Objectives**: Transdermal formulations are widely utilized in the pharmaceutical and cosmetic fields because they enable non-invasive administration and sustained local drug delivery. Conventional ex vivo skin permeation experiments using Franz diffusion cells have limitations in capturing the spatial and temporal dynamics of skin penetration. This study aimed to develop a confocal laser scanning microscopy (CLSM)-based approach to visualize and semi-quantitatively assess the penetration behavior of fluorescent dyes with differing lipophilicities. **Methods**: Four fluorescent dyes with different Log P values—Rhodamine B (Rho-B), Rhodamine 123 (Rho-123), Fluorescein Sodium (Flu-Na), and Nile Red (NR)—were formulated into lotion-based vehicles and applied to excised human abdominal skin. CLSM imaging was performed from 10 min to 240 min post-application. Fluorescence intensities were extracted from depth-resolved regions (R1–R4, 30-μm intervals) to examine penetration kinetics and distribution. **Results**: CLSM imaging demonstrated that Rho-B penetrated through stratum corneum and entered deep into the skin via the hair follicles. Rho-123 and Flu-Na exhibited intercellular and follicular penetration; however, Flu-Na showed only a slight increase in intensity over time; NR showed negligible penetration into the deeper layers. The results of our analysis indicated that moderately lipophilic substances such as Rho-B and Rho-123 diffused deeply into the skin via both transdermal and follicular routes, whereas highly hydrophobic or lipophilic substances remained in the superficial layers. **Conclusions**: The CLSM-based approach enabled spatially and temporally resolved, semi-quantitative evaluation of transdermal penetration in a single, non-destructive experiment. Although restricted to fluorescent probes, this approach provides a practical early-stage screening tool for comparing route-specific and time-dependent penetration behaviors of compounds with different lipophilicities.

## 1. Introduction

Topical formulations have attracted increasing attention in both the pharmaceutical and cosmetic fields because of their advantages, such as bypassing the hepatic first-pass effect, enabling non-invasive and user-friendly administration, and allowing drug concentrations to be sustained in the target area [1,2,3]. In particular, recent developments in drug delivery technologies, including nanoparticles and microneedle systems, have further enhanced their applicability by increasing the efficiency of drug transport to target sites [4,5,6,7].

The permeability of active pharmaceutical ingredients through the skin is known to be influenced not only by their molecular weight and charge but also considerably by their lipophilicity (Log P) [8,9,10]. Log P influences both the extent of penetration and the route of drug permeation, that is, through the intercellular lipid matrix and through trans-appendageal routes such as the hair follicles [10]. In the case of nanoparticle-based formulations, hair follicle targeting has been reported to play a major role, with the particle size having been shown to be correlated with the depth of penetration [11,12].

Ex vivo skin permeation experiments using Franz diffusion cells have long been the gold standard for assessing transdermal drug delivery [13,14,15,16]. This method typically employs excised human or animal (e.g., porcine or rodent) skin or three-dimensionally reconstructed skin models, with the drug concentrations measured in receptor fluids or skin tissue following topical application [13,14]. However, it has notable limitations. The procedures involved, comprising skin trimming and thickness adjustment, formulation application, sample collection (e.g., tape stripping, epidermis isolation, or follicle extraction), and receiver fluid replenishment, are labor-intensive and highly operator-dependent, often resulting in inter-sample variability. Furthermore, when comparing multiple formulations, large sample sizes are required to account for such variability. Additionally, since sampling must be performed destructively at each time point, continuous observation of the same skin area over time is not feasible. As a result, it remains challenging to comprehensively characterize the spatial and temporal aspects, such as the penetration depth, route, and kinetics, of transdermal drug behavior [15,16].

Moreover, early-stage formulation development requires rapid, non-destructive, and low-sample-demand evaluation methods capable of screening multiple candidates under standardized conditions, a need that conventional Franz cell systems do not readily fulfill.

Optical imaging technologies have been increasingly utilized for assessing transdermal drug behaviors to overcome these constraints. These techniques, including confocal laser scanning microscopy (CLSM), confocal Raman microscopy, and multiphoton microscopy, which were originally developed for visualizing cellular behaviors, have recently been applied to the study of the skin permeation behaviors of topical drugs [15,16,17]. Among these, CLSM permits high-resolution, non-destructive visualization of the spatial distribution of fluorescent probes within the skin and has been reported to facilitate the observation of drug diffusion at the skin surface and of drug penetration into the stratum corneum [17,18]. Previous studies have already demonstrated the usefulness of CLSM for elucidating formulation-specific penetration routes. For example, one study visualized the follicular pathway of an emulsion designed for follicular targeting and quantified fluorescences of up to 400 μm below the skin surface [19]. Another report employed time-lapse CLSM to construct three-dimensional images of a microemulsion, thereby tracking the evolving penetration profile of the formulation in real time [20]. Furthermore, particle-size effects (25–100 nm) have also been examined by CLSM, indicating size-dependent migration into furrows and hair openings [21]. Complementary permeation assessments indicate that skin absorption is optimal at Log P = 2–3 [10], whereas efficient follicular targeting is achieved near Log P = 1 [12]. Despite these advances, no prior work has captured the lipophilicity-dependent shift in penetration pathways as a continuous spatial-temporal process.

In this study, we evaluated a CLSM-based ex vivo screening-oriented approach to visualize and compare the skin penetration behaviors of model substances with different lipophilicities in both spatial and temporal dimensions. Specifically, we formulated four lotion-based topical preparations containing fluorescent dyes with varying Log P values from −1.5 to 3.8, applied them to excised human abdominal skin specimens, and assessed their skin penetration profiles using CLSM imaging and fluorescence intensity analysis. The application site was observed at multiple time-points from 10 min after application until 240 min, generating depth-resolved Z-stacks at 30-μm intervals. Fluorescent intensities within each section were extracted to provide a semi-quantitative comparison of the spatial-temporal penetration behaviors across the Log P series. This workflow enables rapid, depth-resolved comparison of multiple candidates under unified conditions, providing a practical tool for early-stage screening in topical and transdermal formulation development. In addition, the Z-stacks were rendered into time-lapse videos, enabling full three-dimensional visualization of the kinetic differences among the four test formulations.

## 2. Materials and Methods

### 2.1. Materials

Rhodamine B (Rho-B), Rhodamine 123 (Rho-123), Nile Red (NR) and Sodium Fluorescein (Flu-Na) were purchased from Sigma-Aldrich (St. Louis, MO, USA) (Table 1). 1× PBS (-) and ethanol were purchased from Wako Pure Chemical (Osaka, Japan), and propylene glycol (PG) was purchased from ADEKA (Tokyo, Japan). Purified water was obtained from a laboratory-grade water purification system (Nikko Chemicals, Tokyo, Japan). Abdominal human skin specimens were purchased from Biopredic International (Rennes, France).

### 2.2. Preparation of Fluorescent Dye-Dissolved Formulations

The fluorescent dyes were dissolved in a mixed solvent consisting of ethanol, PG, and water. The ethanol/PG/water ratio (50/20/30, *w*/*w*/*w*) was selected because this three-component vehicle is widely used in topical formulations, particularly hair-growth preparations [26], and provides sufficient solubility for all four dyes under identical thermodynamic conditions. Using a uniform vehicle composition minimized differences in thermodynamic activity that could influence percutaneous absorption. Each dye was incorporated at 0.002 wt%.

### 2.3. Skin Sample Preparation for Confocal Laser Scanning Microscopic (CLSM) Imaging

Human abdominal skin specimens were purchased from Biopredic international (Rennes, France; specified thickness 200–400 μm) were stored frozen at −20 °C until use. After thawing at 4 °C, each sample was visually inspected to ensure the absence of tearing, peeling, excessive dryness, or localized structural damage, and only specimens with an intact epidermal structure were used. The specimens were cut into 5 mm × 5 mm squares and washed with 1× PBS (-). They were then set on paper soaked in 1× PBS (-), followed by application of 20 μL of each of the fluorescent dye-dissolved formulations. Thereafter, the specimens were mounted onto a 35 mm glass bottom dish and observed by CLSM.

### 2.4. CLSM Imaging

Skin permeation by the fluorescent dyes was observed using the Zeiss LSM 800 confocal laser scanning microscope (Carl Zeiss Microscopy GmbH, Jena, Germany). A 10× objective lens (Plan-Apochromat 10×/0.45 M27) was used for the observations, and two types of lasers were selected as appropriate according to the excitation wavelengths of each fluorescent dye. Each fluorescent dye was evaluated using an independent skin specimen, and only one dye was present per sample; thus, spectral overlap or cross-talk could not occur during CLSM acquisition. The laser intensity was set in the range of 0.2% to 2.0% depending on the fluorescent dye. Table 2 shows the excitation/detection wavelength range and image color of each fluorescent dye. The pixel size of the 3D images was 512 × 512 (8 bits), and the Z-Stack condition was set to 101 slices (Z = 300 μm) every 3 μm. Under the above conditions, time-point images were obtained from 10 min until a maximum of 240 min after the dye application to the skin under a dark-room condition at 20 °C–50 RH%.

### 2.5. Analytical Conditions

Image construction and brightness analysis were performed using Imaris 9.8 (Oxford Instruments KK, Tokyo, Japan). A 3D region was defined (X = 512 pixels [639 μm], Y = 161 pixels [201 μm], and Z = 101 pixels [303 μm]) according to the location of the hair follicle. X was set parallel to the follicle, Y perpendicular, and Z in the depth direction. 3D permeation images of each fluorescent dye were generated using the Surface function. For semi-quantification, the 3D images were divided into four depth regions using the Crop function. A maximum depth of 120 μm from the skin surface was selected, and sections R1- R4 were defined every 30 μm. The selection of 30-μm segmentation was based on (i) the typical stratum corneum thickness of abdominal skin (~10–20 μm) [27,28], ensuring that R1 fully contains this anatomical layer; (ii) natural unevenness of the skin surface and specimen-to-specimen variability, which could lead to misalignment if depth intervals were too fine; and (iii) the need to balance anatomical interpretability with analytical robustness for depth-dependent comparison. For each of the region (R1–R4), fluorescence intensity of each dye was calculated at multiple time points up to 240 min after application (Figure 1). Because one independent human skin specimen was used per fluorescent dye (n = 1 per dye), the temporal and depth-resolved Z-stack measurements represent technical rather than biological replicates. Therefore, fluorescence data were evaluated as relative temporal changes within each dye, and no statistical comparisons across dyes were performed.

The total Intensity Sum for each region was defined as the amount of fluorescent signal, and the rate of change over time was calculated for each dye from 10 min after application. In addition, the abundance ratio of the fluorescence in each region was calculated. The formulae used for these calculations were as follows:Relative change ratio (%) = (Sum intensity in each region at X min/Sum intensity in all regions at X min)/(Sum intensity in each region at 10 min/Sum intensity in all regions at 10 min) × 100(1)Existence ratio of fluorescence (%) = Sum intensity in each region at X min/Sum intensity in all regions at X min × 100(2)

## 3. Results

### 3.1. CLSM Imaging of Rho-B over Time

Figure 2 and Figure 3 show the CLSM images of the skin surface and the deeper layers after the application of Rho-B from 10 to 240 min. In the XY plane, Rho-B (shown in red) gradually dispersed between 10 and 60 min, followed by a moderate increase in overall fluorescence intensity until up to 240 min (Figure 2). In the XZ plane, the fluorescence intensity in the stratum corneum increased from 10 to 60 min, with diffusion into the deeper skin regions via the hair follicles seen up to 240 min (Figure 3). These results indicate that Rho-B initially penetrated into the stratum corneum and subsequently moved into the deeper layers of the skin over time, primarily via follicular pathways.

### 3.2. Semi-Quantitative Analysis of the Fluorescence Intensities

The fluorescence levels at each observation time-point were standardized to the intensity recorded at 10 min post-application (Figure 4). As shown in Figure 1, the observation area was segmented into four depth layers at 30-µm intervals from the skin surface, labeled R1 to R4. The fluorescence intensity in the outermost layer (R1) gradually decreased to 62% after 240 min. Conversely, R2 showed a relatively stable at 86%, although significant increases in values were observed in the deeper layers, R3 and R4. AT 240 min, the fluorescence intensity reached 210% in R3 and 240% in R4.

Figure 5 shows the comparative distributions of the fluorescence intensities from R1 to R4 over time. Initially, more than 80% of the fluorescence was concentrated in the superficial layers (R1 and R2); however, by 240 min, a distinct transition to deeper layers (R3 and R4) had occurred. The intensity in R1 decreased from 28.6% to 17.6%, whilst that in R3 and R4 increased from 12.5% to 26.3% and 4.2% to 9.0%, respectively. These findings indicated that Rho-B initially diffused through the stratum corneum and subsequently penetrated the deeper layers through follicular pathways.

### 3.3. CLSM Images of the Fluorescent Dyes with Different Lipophilicities over Time

Figure 6 and Figure 7 show CLSM images from the XY and XZ planes, respectively, captured 10- and 240-minafter application for the four fluorescent dyes with differing Log P values: Rho-B, Rho-123, Flu-Na, and NR. As previously described, Rho-B showed diffusion through both the stratum corneum and follicular openings and progressively accumulated in deeper regions. Rho-123 showed fluorescence delineating cell membranes at 240 min, signifying intercellular penetration. Furthermore, we noted infiltration through the follicular routes and heightened fluorescence in the deeper layers. Flu-Na exhibited similar intercellular and follicular diffusion; however, the fluorescence intensity did not markedly increase over time. NR produced bright signals in the stratum corneum and hair follicle regions within 10 min, however, its intensity showed no significant enhancement, particularly in the deeper layers, even at 240 min.

### 3.4. Comparison of the Skin Penetration Behaviors of the Four Fluorescent Dyes with Different Lipophilicities

Figure 8 shows the temporal changes in the normalized fluorescence intensity for each dye, with the value at the 10 min time-point used as the reference baseline. For Rho-123, slight reductions were noted in R1 and R2 (to 89% and 96–113%, respectively), whereas substantial increases were noted in R3 and R4, where the values reached 187% and 241%, respectively, at 240 min. Flu-Na exhibited comparable tendencies, albeit with smaller magnitude (R3: 142%, R4: 164%). NR showed negligible variations across R1-R3 (120%, 86%, and 98%, respectively), whereas R4 showed a progressive increase, reaching 147% at 240 min without an initial rapid surge.

Table 3 shows the changes in the regional fluorescence distribution (R1 to R4) from 10 to 240 min for each dye. Rho-B, Rho-123, and Flu-Na showed reductions in R1, alongside concomitant increases in R3 and R4, indicating progressive penetration into deeper layers of the skin. Conversely, NR showed no apparent distributional changes in the deeper layers.

## 4. Discussion

In this study, we established a CSLM-based method as a useful approach to spatially and temporally visualize and semi-quantitatively assess the transdermal penetration behaviors of lotion-based topical formulations containing four fluorescent dyes with different lipophilicities (Rho-B, Rho-123, Flu-Na, and NR). By continued imaging of the same skin specimen for 240 min and segmenting Z-stacks every 30 μm, we were able to track the exact moment a fluorescent dye left the skin surface and penetrated into deeper layers of the skin along with the pathway information and quantify the associated fluorescence gain or loss in each region.

Because optical scattering and absorption attenuate fluorescence signals with depth, particularly beyond ~100–150 μm, the intensities observed in deeper regions (R3 and R4) may be underestimated. For this reason, the present study focused on temporal changes within each depth region, using fluorescence intensity at the initial time point as the reference. This design enabled relative, semi-quantitative evaluation of penetration kinetics without relying on absolute inter-depth comparisons.

Subongkot et al. imaged a follicle-targeted finasteride microemulsion at a single observation point; while the fluorescence was integrated from 0 to 400 μm, no kinetic information or vehicle comparison was performed in that study [19]. Our time-course observations showed that Rho-B first appeared in the follicles at 30 min after the application; the fluorescence in R3 and R4 rose by 2.1-fold and 2.4-fold, respectively. Zou et al. compared nanoparticles of 25–100 nm diameter at a single time-point and reported size-dependent localization, without quantification [21]. In contrast, our study employed continuous imaging over 240 min, enabling a shift from static, qualitative snapshots to a spatio-temporal, semi-quantitative analysis capable of capturing dynamic penetration profiles. The fluorescence intensity in each bin was normalized to the total signal, allowing us to chart the progressive shift in the dye from the surface regions to the deeper layers and to calculate the relative abundance of fluorescence in each compartment. By calculating the time-dependent percentage distribution of Rho-B across the four depth regions (R1–R4), we observed that the fluorescence intensity in the surface zone R1 decreased from 28.6% to 17.6%, whereas that in the deeper zones R3 and R4 increased from 12.5% to 26.3% and 4.2% to 9.0%, respectively, over the observation period. A pronounced redistribution occurred within the first 60 min after application, followed by a more gradual shift up to 240 min, enabling a semi-quantitative characterization of the time-resolved penetration kinetics of Rho-B. Kitaoka et al. formulated a series of water-in-oil microemulsions, and as supplementary evidence of product quality, examined only one of those formulations by CLSM up to 12 h after topical application to human skin; these authors provided clear XY images confirming skin penetration, yet their data remained strictly qualitative and no head-to-head comparison among the different microemulsion prototypes was attempted [20]. In our study, the vehicle type was fixed as a lotion and only the lipophilicity values (Log P) of the four fluorescent dyes varied from −1.5 to 3.8. Z-stacks were acquired for 240 min, segmented into 30-μm bins, and converted to semi-quantitative depth profiles. Dyes with Log P values of around 1 penetrated the deepest and fastest, an optimum that matches the data obtained with Franz cells [10,12]. Thus, continuous CLSM imaging offers a rapid screening approach for identifying penetration optima and route-specific behaviors.

A comparative investigation revealed varied penetration patterns of the four dyes based on their Log P values (Figure 9 and Table 4). Rho-B and Rho-123 showed considerable infiltration into the deeper layers through transdermal pathways (via corneocytes and intercellular gaps, respectively), as well as by follicular diffusion. This was accompanied by a marked reduction in the fluorescence intensity in the superficial layers of the skin (R1 and R2) and an increase in the deeper layers (R3 and R4), especially a rapid rise shortly after the application, followed by more steady accumulation. Conversely, while Flu-Na exhibited similar routes of penetration to Rho-123, the intensity increases in the deeper layers were less pronounced, indicating lower levels of deep-skin permeation. NR showed superficial localization in the stratum corneum, showing only a small overall rise in fluorescence and lacking considerable penetration into the deeper layers, in stark contrast to the case for the other three dyes. However, a slight increase in NR fluorescence was also observed in R4, which may reflect follicular transport. Human sebum contains highly lipophilic components such as triglycerides, was esters, and squalane [29,30], and these could facilitate the movement of hydrophobic molecules like NR along the follicular route.

Given that the stratum corneum of abdominal skin is typically ~10–20 μm in thickness and fully contained within R1 [27,28], the deeper regions (R3 and R4) correspond to the viable epidermis. However, because fluorescence attenuation increases with depth, interpretation of absolute deep-layer intensities remains limited. Accordingly, the profiles presented here should be interpreted as relative, time-dependent trends that reflect redistribution kinetics rather than absolute quantitative permeation.

From a formulation-screening perspective, this CLSM-based approach enables rapid comparison of route-specific and temporal penetration behaviors under a unified vehicle system. The Log P-dependent differences observed—superficial retention, transcellular/intercellular diffusion, and follicular involvement—may support early decision-making in defining target layers and narrowing candidate molecules. However, because this study used excised human skin and single-sample evaluations per dye, factors such as inter-individual variability, repeated-dose accumulation, and vehicle optimization remain beyond its scope. Therefore, this method should be positioned as an early-stage screening tool, with future extension to skin models with exhibiting altered barrier properties and complementary validation techniques such as tissue clearing and MSI. In addition, for future applications involving formulations containing multiple active compounds, the use of single-dye controls and spectral unmixing will be essential to distinguish individual penetration behaviors, and we have noted this as a future direction.

These results confirmed that CLSM can capture both the spatial and temporal penetration dynamics in a single, non-destructive experiment, providing high-resolution, semi-quantitative profiles and 3-D reconstructions of the skin penetration behaviors of topical transdermal preparations (Appendix A). Although the present method is restricted to fluorescent probes and relative quantification, it enables rapid comparison of route-specific and time-dependent penetration behaviors under a unified vehicle system and is therefore useful as an early-stage screening approach. Because CLSM relies on intrinsic fluorescence, label-free alternatives such as confocal Raman microscopy provide chemical specificity but generally offer lower spatial resolution, slower image acquisition, and reduced throughput. Given these considerations, the CLSM-based approach described here should be regarded as a screening tool rather than a stand-alone method for quantitative formulation selection or extrapolation to in vivo performance. Further refinement—such as depth-attenuation correction, validation with tissue-clearing techniques, or AI-assisted image analysis—may enhance its quantitative capability. Moreover, incorporating multiple biological replicates in future studies would allow for the application of statistical procedures (e.g., confidence intervals or smoothing models), thereby improving robustness in evaluating formulation-skin interactions.

## 5. Conclusions

In this study, we successfully utilized a CLSM approach to spatially and temporally visualize and semi-quantitatively assess the skin penetration behaviors of lotion-based topical formulations containing four fluorescent dyes with varying lipophilicities (Rho-B, Rho-123, Flu-Na, and NR). Our approach enabled high-resolution, non-invasive, time-lapse imaging of the skin penetration profiles in both the XY and XZ planes. Using this approach, we demonstrated that moderately lipophilic compounds (Rho-B, Rho-123) not only diffused through the stratum corneum and intercellular spaces but also penetrated into the deeper skin layers via follicular openings. In contrast, highly hydrophilic (Flu-Na) and highly lipophilic (NR) compounds showed limited deep penetration. This spatially and temporally resolved visualization provided comprehensive insights into the depths, pathways, and kinetics of skin permeation, which are not readily accessible by the traditional approach using Franz diffusion cells. Our findings highlighted the efficacy of CLSM as a complementary tool for assessing transdermal delivery systems and analyzing compound behaviors in terms of their physicochemical parameters. Because the present method relies on intrinsic fluorescence and relative quantification, its primary utility lies in early-stage, route-specific screening under standardized vehicle conditions. Although fluorescence-based imaging is essentially relative and subject to optical limitations, its integration with advanced technologies such as tissue clearing, immunostaining, and AI-driven image processing could further enhance its precision and broaden its usefulness. Future incorporation of multiple biological replicates may additionally support more robust quantitative interpretation. Taken together, a CLSM-based evaluation platform presents a possible avenue for standardized, detailed examination of transdermal drug delivery systems throughout their preclinical development.

## Figures and Tables

**Figure 1 pharmaceutics-17-01534-f001:**
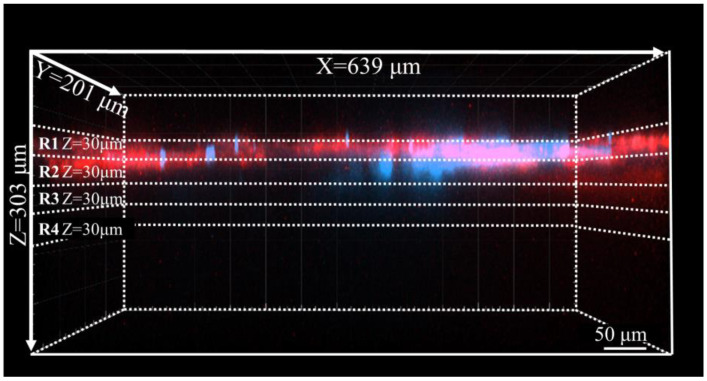
An example of an image observed using CLSM. The X-Y plane corresponds to the skin surface (stratum corneum), and the Z direction corresponds to the depth of the skin, with red indicating a fluorescent dye and blue indicating the hair shafts. The Z direction is divided into four sections of 30 μm each, defined as R1 to R4.

**Figure 2 pharmaceutics-17-01534-f002:**
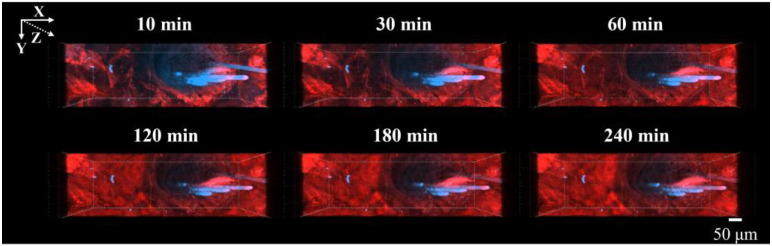
CLSM images of Rho-B from the XY plane. These images were observed from 10 min to 240 min after the application, with red indicating Rho-B and blue indicating the hair shafts.

**Figure 3 pharmaceutics-17-01534-f003:**
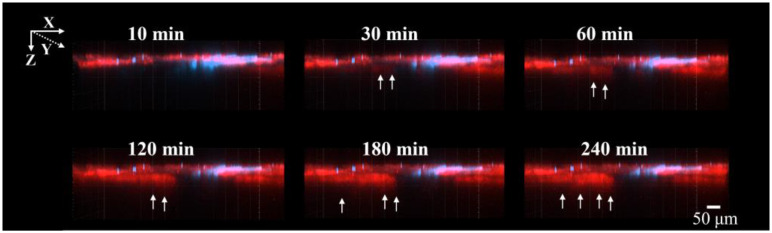
CLSM images of Rho-B from the XZ plane. These images were observed from 10 min to 240 min after the application, with red indicating Rho-B and blue indicating the hair shafts. The white arrows indicate areas where Rho-B penetration into the deep layers of the skin was confirmed.

**Figure 4 pharmaceutics-17-01534-f004:**
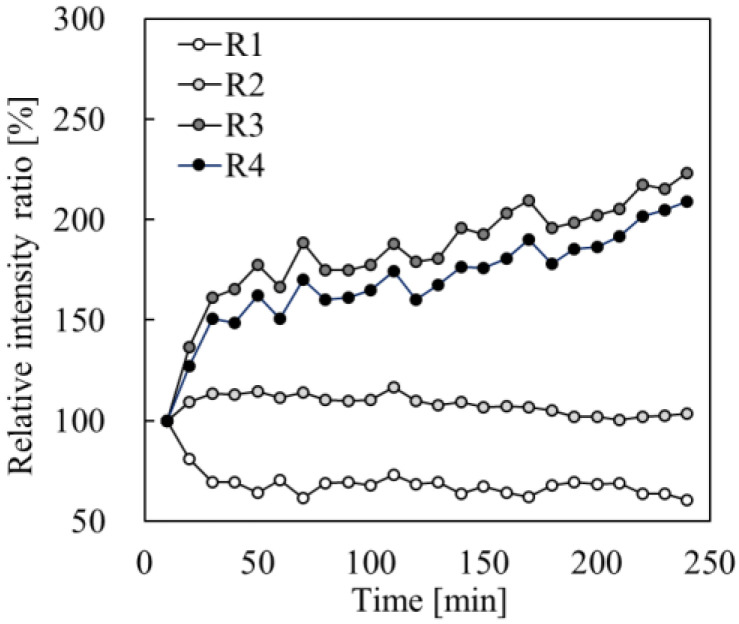
Time-course of changes in the fluorescence intensity ratio of Rho-B in each section (R1–R4), derived from Equation (1). The fluorescence intensity at 10 min after the application was used as the reference value, and the fluorescence intensity ratios at each observation time-point are shown.

**Figure 5 pharmaceutics-17-01534-f005:**
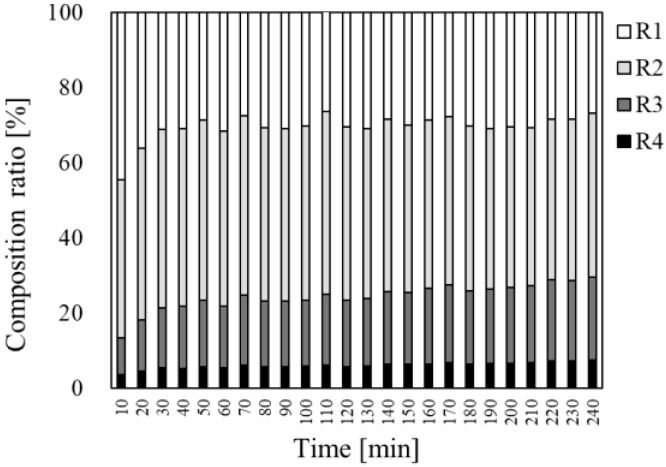
Composition ratios of fluorescence intensity in each section (R1–R4) at each observation point, derived from Equation (2). For each observation point, this figure shows the percentage of the fluorescence intensity in each section relative to the total fluorescence intensity in the entire observation area.

**Figure 6 pharmaceutics-17-01534-f006:**
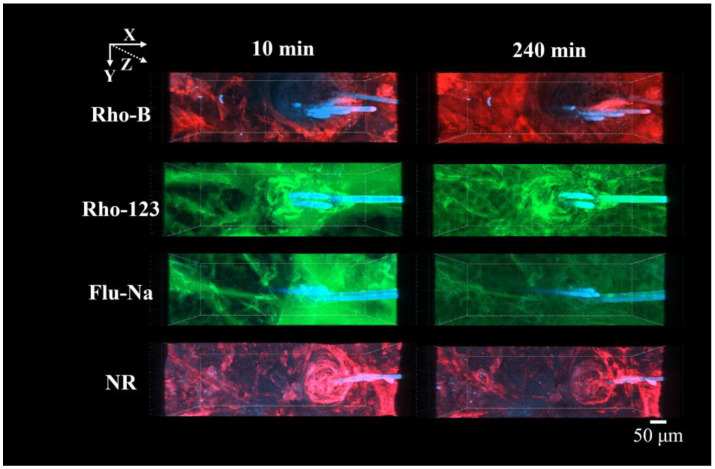
CLSM XY-plane images of each fluorescent dye (Rho-B, Rho-123, Flu-Na, and NR) from 10 min to 240 min after application. Each panel shows a separate specimen containing only one fluorescent dye. Rho-B and NR are displayed in red, whereas Rho-123 and Flu-Na are displayed in green; blue indicates hair shafts. No image contains more than one dye.

**Figure 7 pharmaceutics-17-01534-f007:**
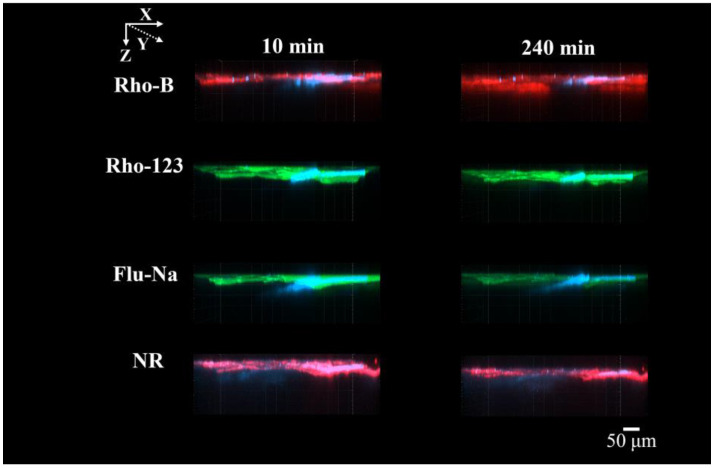
CLSM XZ-plane images of each fluorescent dye (Rho-B, Rho-123, Flu-Na, and NR) from 10 min to 240 min after application. Each panel shows a separate specimen containing only one fluorescent dye. Rho-B and NR are displayed in red, whereas Rho-123 and Flu-Na are displayed in green; blue indicates hair shafts. No image contains more than one dye.

**Figure 8 pharmaceutics-17-01534-f008:**
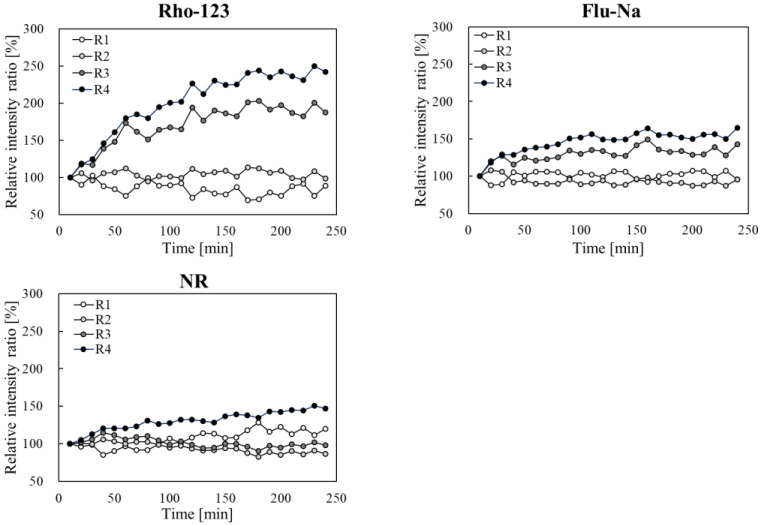
Time-course of changes in the fluorescence intensity ratio of Rho-123, Flu-Na, and NR in each section (R1–R4), derived from Equation (1). The fluorescence intensity at 10 min after the application was used as the reference value, and the fluorescence intensity ratios at each subsequent observation time-point are shown.

**Figure 9 pharmaceutics-17-01534-f009:**
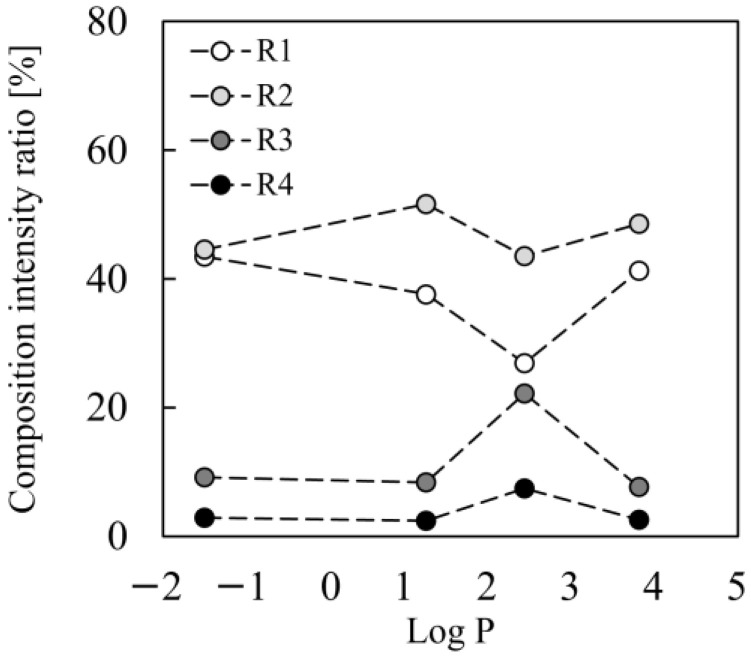
Composition fluorescence intensity ratio of each section (R1–R4) at 240 min after application. The horizontal axis shows the Log P of the fluorescent dyes used in the experiment. Specifically, the values were as follows: Rho-B, 2.4; Rho-123, 1.2; NR, 3.8; and Flu-Na, −1.5.

**Table 1 pharmaceutics-17-01534-t001:** Physical properties of the fluorescent dyes.

	Rho-B	Flu-Na	Rho-123	NR
Structure	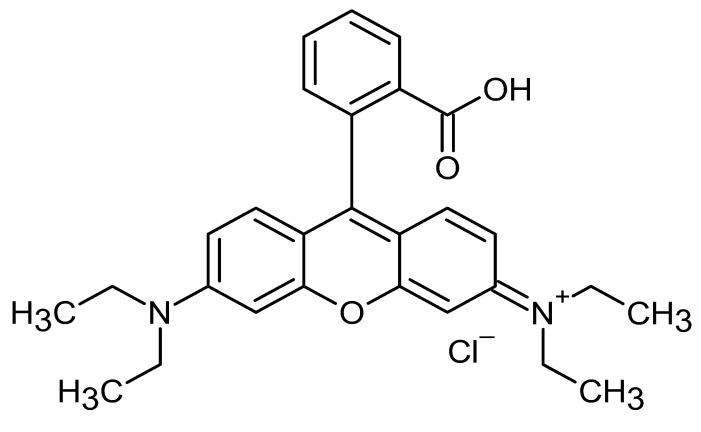	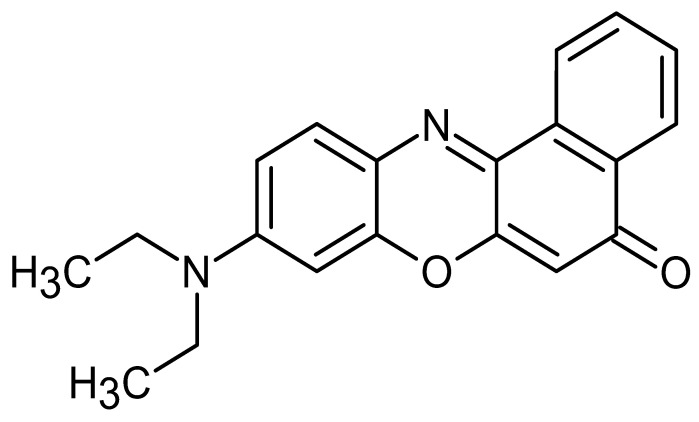	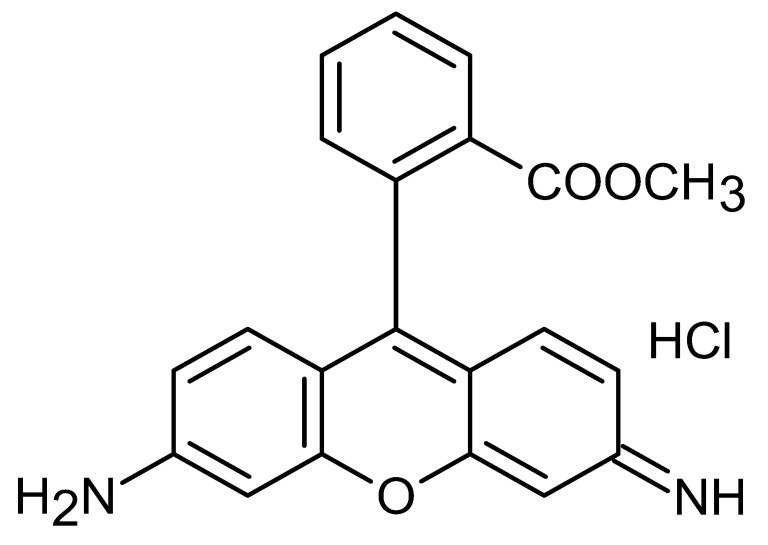	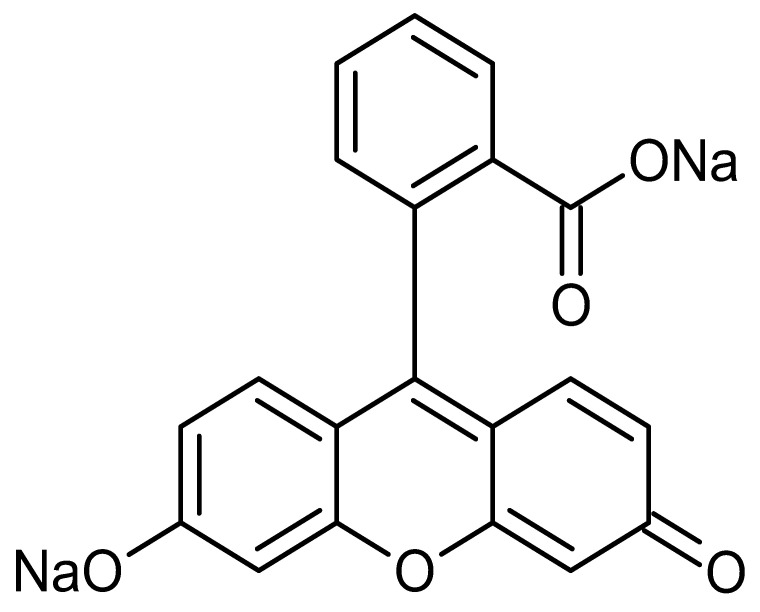
MW	479.0	376.2	380.8	318.4
Log P *	2.4	−1.5	1.2	3.8

*: The reported Log P values in the literature for the four fluorescent dyes vary across the literature. In this study, Log P values of 2.4, −1.5, 1.2, and 3.8 for the four dyes, as reported by Kathuria et al. (2021), Sakai et al. (1997), Chourasia et al. (2011), and Sakata et al. (2014), respectively [22,23,24,25].

**Table 2 pharmaceutics-17-01534-t002:** Fluorescent dye excitation/detection wavelength and image color.

Fluorescent Dye	Excitation Wavelength (nm)	Emission Wavelength (nm)	Image Color
Rho-B	561	570–700	Red
Flu-Na	488	490–640	Green
Rho-123	488	490–600	Green
NR	561	570–700	Red

**Table 3 pharmaceutics-17-01534-t003:** Changes in the composition ratio of fluorescence intensity in each section (R1–R4) from 10 min to 240 min after the application, derived from Equation (2).

Fluorescent Dye	R1	R2	R3	R4
Rho-B	−17.7	1.6	12.2	3.9
Flu-Na	−1.9	−1.9	2.8	1.1
Rho-123	−4.6	−0.7	3.9	1.4
NR	6.9	−7.6	−0.1	0.8

**Table 4 pharmaceutics-17-01534-t004:** Comparison of the skin penetration behaviors of fluorescent dyes with different lipophilicities.

Fluorescent Dye	Flu-Na	Rho-123	Rho-B	NR
Log P	−1.5	1.2	2.4	3.8
Penetration Route	Intercellular gaps Hair Follicle Openings	Intercellular gaps Hair Follicle Openings	Stratum Corneum Hair Follicle Openings	Stratum Corneum
Time-dependent Fluorescence Intensity Profiles in R3 and R4	Initial Rise Gradual Increase	Initial Rise Gradual Increase	Initial Rise Gradual Increase	Gradual Increase
Changes in the Fluorescence Intensity in R3 and R4 Between 10 min and 240 min After Application	Small	Large	Large	Small
Changes in the Ratio of the Fluorescence Intensities in R3 and R4 Between 10 min and 240 min After Application	Large	Large	Very Large	Small

## Data Availability

The original contributions presented in this study are included in the article. Further inquiries can be directed to the corresponding author(s).

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
