# Peer review of "Evaluation of Skin Penetration of Fluorescent Dissolved Formulations Using Confocal Laser Scanning Microscopy"

_pharmaceutics, 2025, doi:10.3390/pharmaceutics17121534_

Round 1

Reviewer 1 Report

Comments and Suggestions for Authors

This manuscript proposes a confocal laser scanning microscopy workflow that tracks the spatio temporal penetration of four fluorescent lotions across excised human abdominal skin. To support publication the study needs calibration, controls and statistics that link intensity to amount, manage optical confounders and generalize beyond a single vehicle and selected fields of view.

  1. Figure 1 bins depth every thirty micrometers regardless of anatomy, so would you consider measuring stratum corneum thickness for each specimen and mapping R1 and R2 to real layers to reduce misassignment?
  2. Table 2 shows different laser powers by dye, yet Figures 6 to 8 are compared across dyes, so can you validate intensity linearity with calibration slides and apply a brightness correction for extinction and quantum yield.
  3. With both Rho 123 and Flu Na excited at four hundred eighty eight nanometers in Table 2, could you document single dye controls and spectral unmixing to exclude bleed through in Figures 6 and 7.
  4. deeper layers scatter more, can you add a depth attenuation correction or a reference channel so the increases in R3 and R4 in Figure 3 and Figure 7 are not underestimated.
  5. The caption for Figure 7 reads as if it shows only Rho B. Please correct the caption and provide a consolidated table of acquisition settings for Figures 2, 3, 6 and 7.
  6. For the summary trends, would you add confidence intervals and the number of stacks per point to Figure 9 and overlay a nonparametric smoother to support the proposed optimum around Log P near one to two.

Reviewer 2 Report

Comments and Suggestions for Authors

The article provides an advanced evaluation of skin penetration using confocal laser scanning microscopy (CLSM) to visualize and semi-quantify the transdermal penetration of fluorescent dissolved formulations with different lipophilicities. Here are detailed reviewer suggestions for improvement.

  1. Suggested to include more comprehensive details on skin specimen handling, including thawing conditions, and confirmation of skin integrity prior to experimentation. This will help ensure reproducibility and minimize variability due to pre-analytical factors.​
  2. Suggested to clarify the rationale behind using the chosen ethanol/propylene glycol/water solvent system for dissolving dyes, especially discussing potential impacts on solubility and skin barrier effects for different Log P candidates.​
  3. Enhance the statistical analysis section by specifying the number of biological replicates and exact statistical methods used for comparing fluorescence intensities between regions and across time points.
  4. Suggested to provide error bars in figures depicting fluorescence intensity and discuss variability between skin samples, which strengthens the credibility of quantitative claims.
  5. Discuss the limitations and potential artifacts when interpreting fluorescence signals at increasing depths, as this is critical when using CLSM for semi-quantitative depth analysis.
  6. Consider adding complementary quantitative analysis such as tissue clearing or mass spectrometry imaging as future work to validate findings from CLSM, especially for compounds with weak fluorescence or complex partitioning behaviors.
  7. Suggested to expand discussion on the translation of these findings to real-world topical and transdermal drug delivery: How would excised skin, formulation vehicle, and dye physicochemical diversity relate to clinical or cosmetic applications?
  8. Please integrate a discussion on inter-individual skin variability (e.g., age, hydration, or disease status), and how these might influence the observed penetration routes and kinetics.
  9. Address more explicitly the study’s limitations regarding the use of excised human abdominal skin and the absence of in vivo or clinical correlation.
  10. At last suggests the plans for expanding the scope to include skin models with different barrier properties, exploring the effect of formulation excipients, or integrating AI-driven image analysis in future studies.

Reviewer 3 Report

Comments and Suggestions for Authors

Dear Authors,

I found the manuscript "Evaluation of Skin Penetration of Fluorescent Dissolved Formulations Using Confocal Laser Scanning Microscopy" interesting, but I have several comments:

  1. Recent scientific literature accounted for more than half of all articles. A total of 25 articles were reviewed: 14 articles (56.0%) are older than 5 years (2025–2020), 4 articles (16.0%) are older than 5–10 years (2019–2014), 5 articles (20.0%) are older than 10 years (2013–2000), and 2 articles (8.0%) were 1992 and 1997. The information provided is the most up-to-date.
  2. I recommend writing "ex vivo skin permeation experiments" instead of "in vitro skin permeation experiments".
  3. What was the thickness (µm) of the skin specimens? Lines 118–124

Reviewer 4 Report

Comments and Suggestions for Authors

Here Y. Oaku et al demonstrated the utilization of confocal laser scanning microscopy (CLSM) approach for spatio-temporal evaluation of formulations for topical skin penetration. The authors tested four fluorescent dyes with different lipophilicity (log P) as model substance for the demo of such approach, with monitoring every 10min interval for upto 240min. 

Overall, the idea seems quite interesting and would be useful for evaluation of various transdermal formulations. I would recommend the acceptance of this article, following sufficient clarifications / revisions below:

  • Is there limitation of Z-depth that can be evaluated through this approach? Does the fluorescence intensity show tendency to reduce as the dye penetrated deeper? Moreover, is there particular reason for division into 4 sections of 30um depth each?

  • Presented CLSM method works well for compounds that are intrinsically fluorescent, particularly on the red/NIR/IR spectrum. Could the authors describe the feasibility of such method for non-fluorescent test substances? Conjugating/tagging fluorophores may alter the physicochemical properties of test substance and thus undesirable. 

  • Formulation composition was defined as 50/20/30% of ethanol/PG/water, with 0.002wt% fluorescent dye. Can the authors explain why such formulation was selected? I believe composition would significantly influence penetration efficiency of the topical formulation

  • For Fig 8 & Table 3 time-course intensity changes, can the authors explain more for the positive trend seen in both R1 & R4 section for Nile Red (NR)? I thought retention at superficial layer should only be reflected with accumulation in R1 or R2 section.

Round 2

Reviewer 1 Report

Comments and Suggestions for Authors

The quality of the revised manuscript has been significantly improved, and I recommend that the revised manuscript be accepted.

Reviewer 2 Report

Comments and Suggestions for Authors

The authors have reflected all the said suggestions and comments, which made the manuscript enhanced with improved readability; Thus, I suggest for further consideration with acceptance.